# Automatic Detection of Olive Tree Canopies for Groves with Thick Plant Cover on the Ground

**DOI:** 10.3390/s22166219

**Published:** 2022-08-19

**Authors:** Sergio Illana Rico, Diego Manuel Martínez Gila, Pablo Cano Marchal, Juan Gómez Ortega

**Affiliations:** 1Robotics, Automation and Computer Vision Group, Electronic and Automation Engineering Department, University of Jaén, 23071 Jaén, Spain; 2Institute for Olive Orchards and Olive Oils, University of Jaén, 23071 Jaén, Spain

**Keywords:** Delaunay triangulation, high-resolution aerial images, multispectral imagery, olive tree canopy, precision agriculture, remote sensing, thick plant cover, UAV, weeds

## Abstract

Marking the tree canopies is an unavoidable step in any study working with high-resolution aerial images taken by a UAV in any fruit tree crop, such as olive trees, as the extraction of pixel features from these canopies is the first step to build the models whose predictions are compared with the ground truth obtained by measurements made with other types of sensors. Marking these canopies manually is an arduous and tedious process that is replaced by automatic methods that rarely work well for groves with a thick plant cover on the ground. This paper develops a standard method for the detection of olive tree canopies from high-resolution aerial images taken by a multispectral camera, regardless of the plant cover density between canopies. The method is based on the relative spatial information between canopies.The planting pattern used by the grower is computed and extrapolated using Delaunay triangulation in order to fuse this knowledge with that previously obtained from spectral information. It is shown that the minimisation of a certain function provides an optimal fit of the parameters that define the marking of the trees, yielding promising results of 77.5% recall and 70.9% precision.

## 1. Introduction

According to the Food and Agriculture Organization of the United Nations (FAO), in the year 2020 there were 12.8 million hectares dedicated to olive trees in the world [1]. Although far from the figures of wheat, the most cultivated crop in the world with 219 million hectares of dedicated land, olive trees stand in the 22nd position out of 161 in the ranking of primary crops. The European Union (EU) represents 40% of the dedicated land with 5.1 million hectares, with Spain accounting for 2.6 million hectares and 51.4% of the European crop area, followed by Italy with 1.1 million hectares and 22.4%. These figures translated into a total world production of 3.3 million tons of olive oil [2], of which 57.9% (1.9 million tons) were produced by the EU. Within the EU, Spain accounted for 1.1 million tons and 58.6% of the European production, with Italy producing 19.1% (0.37 million tons). In turn, the table olive world production was 2.96 million tons, of which 21.9% (0.65 million tons) were produced by Egypt and 26% (0.77 million tons) by the EU. Within the EU, Spain was the largest producer with 0.46 million tons and 58.6% of the European production.

These figures explain the interest in the olive oil and table olives production from the grove to the final production stages in the olive oil factories. Plenty of research has been devoted to the improvement of the quality of olive oil by means of controlling the key variables of the production process [3,4,5]. However, since no chemical or biochemical coadjuvants can be employed to improve the olive oil features, as the production process is carried out by mechanical means exclusively, focusing only on the operations on the olive oil factory means that the best that can be achieved is to preserve the quality properties of the olives arriving at the factory. Therefore, current research trends are starting to focus also on the improvement of the quality and productivity in the stages where the olives are still on the tree.

Regarding this, there are works that study different watering strategies [6,7], their effect on the organoleptic properties of the produced olive oil [8] or how the hydric stress affects the olive trees [9,10]. Other works focus on the early detection of plagues, such *Xylella fastidiosa* [11,12,13] and *Verticillium* [14], or the assessment of the nutritional state of the trees [15,16], while others study the way to evaluate different parameters such as the volume, height and width of the trees [17,18].

Many of these research works share a common factor, namely the acquisition of information on the Earth’s surface using different types of sensors, such as visible spectra, multispectral and thermal cameras, *LiDAR* laser sensors or synthetic-aperture radars, mounted on satellites or unmanned aerial vehicles (UAV). Later, this information is used, in conjunction with novel image processing methods [19], in several tasks such as monitoring the temporal evolution of surface deformations [20] or mapping potential landslide runout zones [21]. This is what is known as remote sensing. When these research approaches are focused on the study of fruit tree groves, such as olive trees, a common necessity is to extract features from the pixels of the images belonging to the tree canopy, since, in order to verify the proposed hypothesis, the experimental results need to be compared with some ground truth obtained from the use of other sensors on the trees.

In order to relate these measurements with the features obtained from the pixels of the tree canopies in the images, it is required to delimitate each of these canopies, which is an arduous and tedious process were it to be performed manually [22]. Therefore, most of the time, authors tend to use more or less standard software tools when the simplicity of the images allows it [11,12,23]. Many works employ spectral values of the images to perform this segmentation, which could bias the features extracted afterwards [24,25]. Only a reduced number of articles explicitly address the problem of identifying the tree canopies as their main topic.

Among them, it is worth highlighting [26], in which the authors use the Gram–Schmidt spectral sharpening fusion method to integrate the panchromatic and multispectral images and train a convolutional neural network (CNN) that manages to detect oil palm trees with up to 98.65% precision and 98.88% recall using images obtained from the Quickbird satellite. A different approach is applied in [27], where the authors use the red and infrared spectral bands to compute the normalised difference vegetation index (NDVI) and apply classic computer vision methods such as thresholding and blob detection, obtaining a relative error of between 0.2% and 20.7%.

The next step to increase the ground sample distance (GSD) would be to perform the data acquisition using airborne sensors mounted on aircrafts. In [28], the authors use principal components analysis to process a single band image derived from the 8 bands the sensor captures. They then apply a two-stage approach with edge detection followed by marker-controlled watershed segmentation on an image of a forest of spruce trees, Douglas firs and subalpine firs. According to the number of labelled trees, a recall of 85.3% is achieved, while the recall computed using the number of labelled pixels yields 75.6%. Ref. [29] employs colour-infrared images of forested regions and applies a fuzzy thresholding algorithm to find the seeds that are used by a region growing algorithm. As result, 73% of trees were correctly found, with a variation from 62% to 81%. Some more examples of the use of CNN for individual tree canopy detection can be found in [30], in which the data acquired by an RGB sensor is merged with the information collected by a *LiDAR*. A previous segmentation of the trees of an open woodland of live oak, blue oak and foothill pine is performed using the *LiDAR* information, so later the CNN can be trained with each of these Region Of Interest (ROI). This model has an average tree canopy recall of 0.69 with a precision of 0.61. In [31], citrus and other crop trees are detected from UAV images using a simple CNN algorithm, followed by a classification refinement using superpixels derived from a simple linear iterative clustering algorithm, achieving 94.59% precision and 97.94% recall.

When high-resolution images are captured with UAVs, contrary to the methodology followed in this article, most of them extract structured 3D information using *LiDAR* sensors or applying photogrammetric analysis. These methods can considerably improve the results obtained, especially when it is necessary to differentiate pixels that are spectrally similar but that are at different heights from the ground (tree canopies and weeds). Even so, they have some disadvantages that are discussed in Section 4. For example, [32] uses UAV LiDAR data for individual tree detection in subtropical mixed broadleaf forests in urban scenes. This method improves on the popular local maximum filter by removing those LMs caused by surface irregularities contained in the canopy height model, and obtains an F-score between 73.7% and 93.2%, depending on how irregular the distribution of trees or crown size is. In [33], RGB images of a maize plantation at the seedling stage are combined with *LiDAR* data captured by a UAV. The maize seedlings extracted from the images serve as seeds for the fuzzy C-means clustering algorithm used to segment individual maize plants. The results revealed an accuracy with R${}^{2}$ greater than 0.95, a mean square error (RMSE) of 3.04–4.55 cm and a mean absolute percentage error of 0.91–3.75%. Other examples of articles using photogrammetry to detect and extract trees from high-resolution UAV RGB images are [34] for citrus trees, [35] for peach trees, [36] for chestnut trees and [37] for papaya trees. The first uses sequential thresholding, Canny edge detection and circular Hough transform algorithms on the Digital Surface Model (DSM), obtaining accuracies that exceeded 80%. The second uses an adaptive threshold and marker-controlled watershed segmentation in the DSM to measure the canopy width and canopy projection area, achieving an RMSE of 0.08–0.47 m and 3.87–4.96 m${}^{2}$, respectively. The third applies a segmentation stage based on the computation of vegetation indices combined with the canopy height model to extract the candidate tree canopies. The next stage receives these tree canopies to divide those that are greater than a threshold. The last stage extracts the desired features. The segmentation accuracy of this method is above 95%. Finally, in the fourth article, the authors make an improvement with regard to the existing scale-space filtering by applying a Lab colour transformation to reduce overdetection problems associated with the original luminance image. The achieved F-score is larger than 0.94.

As concluded in [38] and evidenced in the previous paragraphs, no algorithm is optimal for all types of images and plant types. For this reason and for clarity, articles related specifically to olive trees and how to detect their canopies have been compared in the Section 3. Although these articles are closely related to the issue at hand, none of them explicitly deals with images from groves with thick plant cover on the ground without using three-dimensional data. Finally, to complement this literature review, mention should also be made of articles dealing with detecting weeds in crops of herbaceous plants such as maize [39], sugar beet [40], sunflower and cotton [41,42] or bean and spinach [43].

The objective of this paper is to develop a method to segment olive tree canopies from high-resolution aerial images that contain information of the visible spectra, specifically, the red, green and blue bands that can cope with high levels of plant cover in the ground between canopies.

The key idea of the approach is to employ not just the spectral information contained in the images but to also consider the relative distance between canopies, computing and extrapolating the planting pattern used by the grower and fusing this information with the spectral data. This paper shows that the minimisation of a certain function provides an optimal fit of the parameters that define the marking of the trees, yielding promising results, without the need to resort to deep learning methods that are difficult to interpret.

The structure of the paper is as follows: Section 2 presents a general diagram divided into blocks that explains the workflow followed to achieve the results of this research. This section is made up of subsections that correspond to each of the blocks in the diagram. Although the methodology followed is explained in detail in all of them, the first block Section 2.1 also focuses on the materials used during data capture and the way in which it was carried out. In Section 3, first, the quality metrics of the results obtained after the application of the developed model are objectively shown, and these numbers are analyzed together with an explanation of the possible causes of the model’s failures. Second, the execution times of the building blocks of the model are computed, both for the trainings and for the predictions of the model. Third, a comparison is made with other articles related specifically to olive trees and how to detect their canopies. Finally, in Section 4, the objective of the article is expanded and its usefulness and advantages compared with other methodologies are explained. Possible improvements are also discussed and future work is anticipated.

## 2. Materials and Methods

The workflow of the method proposed in this paper is presented in Figure 1 and shows the different materials and methods used, as well as the information transference between the different parts that compose it. This workflow can be divided into two blocks: the *Data Preprocessing (DP)* of the images taken in the groves and the *Olive Tree Canopy Detection Model (OTCD)*, which is composed of three submodels, namely, the *Vegetation Classification (VC)*, the *Olive Tree Canopy Estimation (OTCE)* and the *Olive Tree Canopy Marking (OTCM)*.

The first block *DP* deals with the transformation and separation of the original raw data captured by the sensors to the format required for the input of the model, specifically: multispectral images, metadata associated with these multispectral images and pixels that are labelled to their corresponding class.

This preprocessing task, together with the training of the two first submodels of the *OTCD* block, is performed beforehand—in Figure 1 they are shown with a dashed line. This way, the time required to perform these task does not influence the time required to detect the tree canopies of a new image.

The second block *OTCD* deals with the prediction of the coordinates of each of the tree canopies included in the new images provided as inputs.

### 2.1. Multispectral Captures

The data employed in this work were gathered by the following sensors mounted on a *DJI Matrice 600* UAV:A *Micasense RedEdge-M* multispectral camera, capable of capturing 12 bit images with a 1280 by 960 pixel resolution in five bands of the electromagnetic spectrum. These bands are blue (475 nm), green (560 nm), red (668 nm), near infrared (840 nm) and red-edge (717 nm).A sunlight sensor *Micasense Downwelling Light Sensor (DLS) 1*, capable of measuring the ambient light for each of the five bands of the camera *Micasense RedEdge-M*.A global positioning system (GPS) *Ublox M8N*.

These sensors are completed with a calibrated reflectance panel (CRP) *Micasense RP04-1826404-SC* that takes images of before and after each flight by the UAV in order to be used for the radiometric correction process. The calibrated reflectance values for the specific panel used in this work are 49.1%, 49.3%, 49.4%, 49.3% and 49.4% for the blue, green, red, near-infrared and red-edge bands, respectively.

Each multispectral capture stores the images of each of the five bands in a disk in *.tif* format, together with the metadata provided by the DLS and the GPS.

As commented in the Introduction, the main objective of this paper is to detect olive tree canopies when there is a large amount of plant cover in the ground, so that traditional segmentation methods fail to provide good results. This way, a set of 18 captures (Table 1) with a large amount of plant cover in the ground, were taken from an olive grove in the town of Diezma, province of Granada, in Andalucia, Southern Spain. This is an olive grove of trees of Picual cultivar variety with 29.14 ha of extension, showing a traditional arrangement of the trees. Six flight campaigns were carried out from October 2018 to November 2019. Figure 2 depicts an example of the capture as taken by the multispectral camera, depicting an image for each of the five bands previously mentioned. A thick plant cover can be seen between the olive trees.

All these captures, together with the metadata, were used as inputs for the *Radiometric Correction and Band Alignment (RCBA)*, the manual labelling and the first two submodels of the olive tree canopy detection model.

### 2.2. Radiometric Correction and Band Alignment (RCBA)

Once the multispectral captures are available as inputs to the *RCBA*, the first step is to transform the digital numbers (DN) of the images obtained by the camera sensor to radiance values first and to reflectance values afterwards. This correction, which is carried out for each band, allows the values subsequently employed to be independent of the flight, date, time of the day and climatic conditions, so that different captures can be compared in the same reference frame. Posteriorly, before finishing the *RCBA*, the reflectance images of each band need to be aligned, since the five sensors of the camera have a slight offset from each other.

The computation of the metadata that are obtained from the *RCBA* is computionally expensive, and that is the reason why it is performed beforehand. The results are not applied directly to the multispectral captures but during the training and prediction phase of the models, since the application of the these values is almost instantaneous. This advocates for storing the multispectral captures with these metadata, since it eliminates the need to store all the corrected and aligned images.

For clarity, in the following discussions the dependency of the equations with the wavelength is omitted, but it must be noted that each computation needs to be carried out for each of the five spectral bands captured by the camera.

#### 2.2.1. Raw Images to Radiance Images Conversion

The conversion of the DN into radiance values, $L_{r}$, is carried out using Equation (Equation 1), as recommended by the camera manufacturer, to each of the pixels *x* and *y*, for each of the 5 wavelengths, λ.
(1)$$L_{r}\left(x,y\right)=\frac{a_{1}}{g\cdot t_{e}}\cdot V\left(x,y\right)\cdot R\left(y\right)\cdot \frac{DN\left(x,y\right)-DN_{BL}}{DN_{MAX}}$$

In this equation, DN are the raw digital numbers obtained by the camera sensor and $DN_{BL}$ is the average of the raw values of all the covered pixels of the sensor, whose objective is to measure the small amount of signal captured independently of the incident light. The difference between these two values is normalised dividing by $DN_{MAX}$, which is the maximum digital number achievable, equal to the maximum bit depth of the stored images minus 1. Although the camera captures 12 bit images, they are stored in 16 bits *.tif* format, so the value of $DN_{MAX}$ is $2^{16}-1$. The terms $a_{1}$, *g* and $t_{e}$ refer to the first radiometric calibration coefficient, the gain and the exposure time of the camera, respectively.

The correction of the decrease in the light captured by the sensor from its top to the bottom, R(y), is given by Equation (Equation 2),
(2)$$R\left(y\right)=\frac{1}{1+\left(a_{2}/t_{e}\right)\cdot y-a_{3}\cdot y},$$
where $a_{2}$ and $a_{3}$ are the last two radiometric calibration coefficients.

The vignetting correcting function, V(x,y), is obtained according to Equations (Equation 3)–(Equation 5),
(3)$$V\left(x,y\right)=\frac{1}{k(x,y)},$$
(4)$$k\left(x,y\right)=1+k_{0}\cdot r\left(x,y\right)+k_{1}\cdot r{\left(x,y\right)}^{2}+k_{2}\cdot r{\left(x,y\right)}^{3}+k_{3}\cdot r{\left(x,y\right)}^{4}+k_{4}\cdot r{\left(x,y\right)}^{5}+k_{5}\cdot r{\left(x,y\right)}^{6},$$
(5)$$r\left(x,y\right)=\sqrt{{\left(x-c_{x}\right)}^{2}+{\left(y-c_{y}\right)}^{2}},$$
where *r* is the distance of each pixel to the centre of the vignette ($c_{x}$, $c_{y}$) and $k_{0-5}$ are the polynomial correction factors.

The methods used to obtain Equations (Equation 1)–(Equation 5) have not been shared by the manufacturer Micasense, but the values can be extracted from the metadata tags embedded in the images. They are shown in Table 2.

#### 2.2.2. Radiance Images to Reflectance Images Conversion

The reflectance is defined as the ratio between the reflected and incoming radiant fluxes. Due to energy conservation considerations, this ratio should always lie between 0 and 1. The most general formulation is given using the bidirectional reflectance distribution function (BRDF), denoted by $f_{r}$ [44], as: (6)$$\rho \left(\omega_{i};\omega_{r};L_{i}\right)=\frac{d\phi_{r}\left(\theta_{i},\phi_{i};\theta_{r},\phi_{r}\right)}{d\phi_{i}\left(\theta_{i},\phi_{i}\right)}=\frac{\int_{\omega_{r}}\int_{\omega_{i}}f_{r}\left(\theta_{i},\phi_{i};\theta_{r},\phi_{r}\right)\cdot L_{i}\left(\theta_{i},\phi_{i}\right)\cdot d\Omega_{i}\cdot d\Omega_{r}}{\int_{\omega_{i}}L_{i}\left(\theta_{i},\phi_{i}\right)\cdot d\Omega_{i}},$$
where the subindex *i* refers to incoming magnitudes, and *r* refers to reflected magnitudes. The geometric parameters *w*, θ, ϕ and Ω are the solid angles, azimuth angles, zenith angles and projected solid angles, respectively.

The BRDF is used to describe the dispersion of a ray of incident light on a surface from an incoming direction towards another outgoing direction, and it is considered an intrinsic property of the surface. Its definition is
(7)$$f_{r}\left(\theta_{i},\phi_{i};\theta_{r},\phi_{r}\right)=\frac{dL_{r}\left(\theta_{i},\phi_{i};\theta_{r},\phi_{r};E_{i}\right)}{dE_{i}\left(\theta_{i},\phi_{i}\right)},$$
here, $dL_{r}$ is the infinitesimal reflected radiance and $dE_{i}$ is the infinitesimal incoming irradiance. Given its infinitesimal nature, BRDF is only useful for conceptually understanding other related magnitudes, but it cannot be measured directly, since the raylights do not include any radiant flux. What is measured by the sensors in the multispectral camera and the DLS is the hemispherical–conical reflectance, which is used to obtain the hemispherical–conical reflectance factor (HCRF). The incoming irradiance is hemispheric since, besides the direct solar component, it also considers diffuse components coming from every direction. The reflected radiance is conic due to the fact that each pixel of the camera fills a certain solid angle equal to the instantaneous field of view (IFOV). This IFOV is so small (0.03${}^{\circ }$) that the measurement of the reflected radiance can be considered directional and not conic, so the HCRF could be approximated using the hemispheric–directional reflectance factor (HDRF).

The HDRF is obtained as the ratio between the radiant flux that is reflected by the surface and the radiant flux that an ideal surface (lossless), and a perfectly diffuse (lambertian) would reflect under the same geometric and luminance conditions, since this would be equal to the incoming flux to this type of surface. This parameter could have values between 0 and +∞, although it usually lies below 1 if the reflections are not specular or close to being so.
(8)$$\mathrm{HDRF}=\frac{d\phi_{r}\left(2\pi ;\theta_{r},\phi_{r}\right)}{d\phi_{i}\left(2\pi \right)}=\frac{\cos\theta_{r}\cdot dL_{r}\left(2\pi ;\theta_{r},\phi_{r}\right)\cdot \sin\theta_{r}\cdot d\theta_{r}\cdot d\phi_{r}\cdot dA}{\cos\theta_{r}\cdot dL_{r}^{id}\left(2\pi \right)\cdot \sin\theta_{r}\cdot d\theta_{r}\cdot d\phi_{r}\cdot dA}=\frac{dL_{r}\left(2\pi ;\theta_{r},\phi_{r}\right)\cdot dE_{i}\left(2\pi \right)}{dL_{r}^{id}\left(2\pi \right)\cdot dE_{i}\left(2\pi \right)},$$
since the ideal diffuse reflectance is equal to 1/π,
(9)$$\mathrm{HDRF}=\frac{\pi \cdot dL_{r}\left(2\pi ;\theta_{r},\phi_{r}\right)}{dE_{i}\left(2\pi \right)}.$$

The measurements made by the camera are already integrated through the whole solid angle corresponding to each pixel, as are the measurements of the DLS through the whole upper semisphere, so they can be used to compute the HDRF as: (10)$$\mathrm{HDRF}=\frac{\pi \cdot L_{r}}{E_{i}}=\frac{\pi \cdot L_{r}}{E_{dir}+E_{dif}},$$
here, $L_{r}$ is the radiance reflected by the ground and vegetation surfaces, which is equal to the radiance measured by each pixel of the multispectral camera after it is corrected using Equation (Equation 1), and $E_{i}$ is the incoming irradiance to these same surfaces. This irradiance is divided into the direct irradiance that is perpendicular to the surface, $E_{dir}$, and the diffuse irradiance coming from every direction, $E_{dif}$. The former is computed using Equation (Equation 11), where $E_{s}$ is the same direct irradiance but measured in the direction of the sun, $\theta_{i}$ is the angle between the sun and the horizon, α is the angle between the sun and perpendicular to the irradiance sensor, $E_{dls}$ is the raw measurement of the irradiance sensor, $c_{f}$ is the coefficient applied to account for the reflected radiance that the sensor cannot measure due to the Fresnel effect (the manufacturer provides a value of 0.9057) and $w_{dif}$ is the percentage of diffuse radiation with respect ot the total radiation (it has a value of 0.167 when the sky is clear and the sun is at its zenith). The computation of the latter is carried out using Equation (Equation 12).
(11)$$E_{dir}=E_{s}\cdot \sin\theta_{i}=\frac{E_{dls}/c_{f}}{w_{dif}+\cos\alpha }\cdot \sin\theta_{i}$$
(12)$$E_{dif}=w_{dif}\cdot E_{s}$$

The captures taken from the CRP before and after each flight also need to be taken into account. These captures made on the ground are used to compute a correction factor for each aerial capture, $f_{CRP}$, which is applied to its correspondent incoming irradiance, $E_{i}$. This factor is computed by assigning to each capture of each flight an interpolated irradiance and radiance values (using the timestamps) between the irradiance and radiance measured values from their previous and posterior CRP captures and afterwards applying Equation (Equation 13).
(13)$$f_{CRP}=\frac{\pi \cdot L_{r,CRP}^{interpolated}}{E_{i,CRP}^{interpolated}\cdot {\mathrm{HDRF}}_{CRP}^{calibrated}}$$

Therefore, in this paper, the terms reflectance and reflectance image refer to the computed values of the corrected HDRF.
(14)$${\mathrm{HDRF}}_{corr}=\frac{\pi \cdot L_{r}}{E_{i}\cdot f_{CRP}}$$

#### 2.2.3. Band Alignment

Since the Micasense camera is composed of five image sensors with their corresponding optics, independently mounted at some distance between each other, it is required to perform an alignment of the images that were captured by each of them. This procedure allows to link the reflectance value of each pixel in the green band with the corresponding values of the rest of the pixels that reside on the same spatial coordinate in the rest of the bands.

To perform this alignment, the camera library provides a method called *align_capture*, based on the algorithm *Enhanced Correlation Coefficient Maximisation* [45]. The results of this alignment are part of the generated metadata.

### 2.3. Manual Labelling

There is a large variety of software with different types of licenses that can be used for the manual labelling of the images. These are typically web or desktop applications that allow to assign a label to each image or to certain zones within it, usually delimited using rectangles, polygons, or more complex methods that require computer vision or machine learning algorithms. In any case, the labels are always required to be supervised by a human if they are to be used as references for the evaluation of other algorithms.

For the manual labeling of the images in this work the software needs to be able to work with multispectral images, and since there are no applications in the market that can handle this feature [46], a specific application written in *Python* using the library *Matplotlib* was developed (Figure 3).

This application receives as inputs the aerial multispectral captures of the olive groves together with the metadata computed by the *RCBA*, in order to present already corrected and aligned images to the user. Each delimited region of the image can be labelled as one of four possible labels (olive tree, ground, weed or shadow).

Although each labelled pixel is a vector of six components (five band values and the corresponding class), for the rest of the processes only the red, green and blue bands have been taken into account. The reason is that the results obtained using just these three bands are very promising, and this enables the method to be used with simple visible spectra cameras as well. Additionally, these three reflectance values are transformed to the *CIELAB* colour space, so that the colour values are more perceptibly linear, meaning that if the human eye detects two colours as similar, the coordinates in the *CIELAB* colour space are also close together. Conversely, if two colours are perceived by the human eye as different, the coordinates are far apart. This conversion was carried out using 32 bits floats to take into account that the reflectance takes values between 0 and +∞.

### 2.4. Vegetation Classification (VC)

To be able to detect the olive canopies, the first step is to classify each of the pixels of the images to determine which of those correspond to the olive class or, at least, which of those have a high probability of belonging to that class.

Since during the manual labelling process, only positive cases of pixels for each class are labelled, that is, there is no specific label for pixels not belonging to any of the classes and, furthermore, not every pixel in the image is labelled, any classification algorithm could only discern between these four classes. During the prediction stage, if the algorithm is given all the pixels in an image to assign a class to them, it would be forced to assign one of the four possible classes even to pixels that do not belong to those, generating a large amount of false positives.

A possible solution is to include all the nonlabelled pixels into a fifth class called *other* to perform this classification, but this approach conveys two problems that prevented its use. The first one is that, due to the tedious nature of the manual labelling process, a label is assigned only to pixels that are almost 100% sure to belong to a class, according at the criteria of the person that is performing this manual operation. This reduces the fatigue during the process but provokes that the class other is filled with many pixels that actually belong to one of the other classes, potentially confusing the algorithm enough to impair a proper performance. The second problem is that the pixels classified as other are very heterogeneous due to their multiple origins (roads, cars, rocks, buildings, other types of vegetation, etc.), thus having very disperse values in the feature space and complicating the computation of classification boundaries.

Because of the above reasons, the workflow includes the *VC* block, which represents an initial filtering of the pixels to select only those marked as vegetation (olive tree or weed). The task of discerning between these two classes, which is much more difficult, is left for a later stage.

This vegetation classification is implemented using a one-class classification algorithm (OCC) known as *local outlier factor* [47], which is an unsupervised method to detect atypical values computing the local pixel density deviation with respect to its neighbours. In this case, the algorithm is used to predict if each new pixel belongs to the vegetation class or not, since during the training all the pixels received are not atypical.

In addition to the labelled pixels, the algorithm requires the definition of two parameters whose values are not too relevant for the posterior stages. The contamination value has been chosen to have the maximum possible value (0.5) in order to assure that each pixel labelled as vegetation is effectively vegetation, although that means that pixels that could be classified as such are lost. In turn, the number of neighbours evaluated for each pixel is fixed at a high enough value so that the subsequent steps perform adequately without excessively increasing the prediction time (100).

The output of this process is a vegetation mask (Figure 4) for each capture that represents the zone that, with high probability, is selected as olives or weed in the subsequent stages.

### 2.5. Olive Tree Canopy Estimation (OTCE)

The *OTCE* is based on a decision tree trained with the pixels labelled as olive tree and weed. At prediction time, this tree receives as inputs only the pixels selected by the vegetation mask, in order to classify them as olive tree or weed. However, the results obtained for the olive tree class are not acceptable, nor were they better when different classification algorithms were evaluated. This led to the conclusion that just the spectral information of each pixel independently is not enough to obtain good results, so techniques that take into account the spatial distribution of the pixels in the image were explored.

Nonetheless, this decision provides useful information: the probability image (Figure 5), an image where each pixel of the vegetation mask receives a value between 0 and 1 according to the probability of it belonging to the olive tree class. This probability is computed by dividing the number of pixels classified as olive tree between the total number of pixels inside each leaf of the classification tree.

### 2.6. Olive Tree Canopy Marking (OTCM)

This block is the most important of the whole canopy detection algorithm and is composed of several connected steps, so that the output of one block is the input to the next.

The clipping and normalisation step receives as input the probability image that, basically, is an image with just one channel where the pixel values represent the probability of their belonging to the olive tree class. In this step, the value of all the pixels below a threshold parameter are assigned 0, and the rest of the values are rescaled to work with all the bit depth of the range. This step, therefore, removes those pixels with a probability of belonging to olive tree class below this specified threshold, which is a parameter that needs to be optimized automatically for each image.

The noise-filtering step receives an image where all nonzero pixels have a very high probability of belonging to olive tree class, but they still keep a relative probability information between them (Figure 6a), and applies a simple median filter with the smallest kernel possible (3 × 3) in order to remove isolated pixels that are most likely noise (Figure 6b).

The next step computes the density for each pixel of the probability values, that is, each pixel is assigned the mean of the probability values of the neighbouring pixels and itself, using a mean filter with circular kernel. This steps allows to homogenise to high values the probabilities of zones with clusters of pixels showing high probability, and to low values areas where there are isolated pixels with low probability. The result is an image where the canopies are homogeneously highlighted and isolated areas have been removed (Figure 6c). The parameter optimisation phase carefully selects the size of the kernel for this step.

The last parameter that has to be optimised in the *OTCM* block is the segmentation threshold that, basically, is the threshold that is used to binarise the image during the segmentation step (Figure 6d). All that is left is to search for the contours of this binary image and obtain a list to extract the centroids (Figure 6e).

This sequence of steps allows to achieve the objective of obtaining the correct coordinates of the olive canopies in each image, but, in order to obtain good results, the three parameters previously mentioned need to be chosen appropriately, as they strongly influence the final result, and the optimum value varies notably between captures.

This is the reason to add the parameter optimisation block, whose task is to search for the most adequate values for the probability threshold *p*, the kernel size *k* and the segmentation threshold *s* for each probability image that it receives as input. In order to do this, a brute force method is applied, where the different steps presented above are performed repeatedly for each parameter combination taken from a regular discretisation of their respective domains. For this work, the set *P*, considered for the probability threshold, contains 19 values, starting at 0.05 and reaching 0.95 with increments of size 0.05. The set *K*, related to the kernel size, contains 6 values from 5 to 30 with increments of size 5; finally, the set *S* contains 50 values for the segmentation threshold from 5 to 250 with increments of 5. These sets provide a total of 5700 iterations. In each of these iterations, the Delaunay triangulation (Figure 7) is computed using the coordinates of the olive canopies found (x,y) and the variation coefficient ($C_{v}$) of the set of all of the triangle side length $\left\{l_{1},l_{2},\cdots ,l_{n}\right\}$, removing those that are on the edge (and belong to just one triangle) so that the computation of the ratio between the radius of the inscribed circumference of the adjacent triangle and that of the circumscribed circumference in itself is below 0.1. In order to not leave holes in the triangulation, this procedure is performed recursively until there are no sides that do not fulfill the conditions.

This procedure can be defined in each capture by means of the discrete function *F* as: (15)$$C_{v}=\left\{F\left(p,k,s\right)|\left(p,k,s\right)\subset \left(P\times K\times S\right)\right\},$$
whose value, for each combination of *p*, *k* and *s*, is: (16)$$C_{v}=\frac{\sigma }{\bar{l}}=\frac{\sqrt{N\cdot \sum_{n=1}^{N}{\left(l_{n}-\bar{l}\right)}^{2}}}{\sum_{n=1}^{N}l_{n}},$$
with
(17)$$l_{n}=\sqrt{{\left(x_{i}-x_{j}\right)}^{2}+{\left(y_{i}-y_{j}\right)}^{2}},$$
here, *i* and *j* refer to the indexes of each end of each side of the Delaunay triangulation and *n* to the side itself. A graphical representation in $R^{3}$ of this discrete function can be visualised if any of the three parameters *p*, *k* or *s* is fixed. For instance, Figure 8a,d shows the function for p=0.9, Figure 8b,e depicts the function for k=25 and Figure 8c,f represents it for s=25. These values are the optimum values computed posteriorly by the model for capture 0.

Figure 8 shows that there are no computed values of the *F* function for certain combinations of parameters. This is due to the fact that the algorithm filters the results where it cannot compute the Delaunay triangulations, for instance, if the number of detected canopies is fewer than three. It can also be observed that the values that are close to the limits of the domain fluctuate much more between iterations than the central values, without following any clear trend. This happens because, for these parameter combinations, almost no canopies are found if the values of *p* and *s* are too large or too small (either all the pixels are removed if the values are too high, or very large contours filling almost the whole image are found if the values are too low), or the number of detected canopies is way too large if the values of *k* are too low (all the noise is considered as valid regions). This happens in all the images considered in this work and, in order to solve this issue, we consider only the values of $C_{v}$ that have a number of detected canopies within ±40% of the median number of detected trees for all iterations. That is, a new function *G* that returns the number of detected canopies is defined in Equation (Equation 18), along with the median of the set of values $\left\{t_{1},t_{2},\cdots ,t_{c}\right\}$ computed by *G* in Equation (Equation 19): (18)T={G(p,k,s)∣(p,k,s)⊂(P×K×S)},
(19)$$M=\left\{\begin{array}{cc} t_{(c+1)/2} & \mathrm{if}\;c\;\mathrm{is}\;\mathrm{odd}\\ \frac{t_{\left(c/2\right)}+t_{(c/2)+1}}{2} & \mathrm{if}\;c\;\mathrm{is}\;\mathrm{even} \end{array}\right.,$$
and both are used to redefine the function *F* in Equation (Equation 20): (20)$$C_{v}=\left\{F\left(p,k,s\right)|\left(p,k,s\right)\subset \left(P\times K\times S\right)\;\cap \;G\left(p,k,s\right)\in \left(0.6M,1.4M\right)\right\},$$

The last step is reduced to an optimisation problem of the redefined function *F* to find the values of *p*, *k* and *s* and minimise it in Equation (Equation 21).
(21)$$\left(p_{min},k_{min},s_{min}\right)=\arg\;\min_{p,k,s}\left(C_{v}\right)$$

The graphical representation of the redefined function *F* is depicted in Figure 9, together with the minimum value of the variation coefficient marked. Many values have been removed from those in Figure 8 due to this redefinition.

This methodology, besides using the spectral information of the pixels to detect the tree canopies, includes spatial variables related to pattern detection in images, specifically, the optimum iteration is that whose distribution of canopies provides the least variability, that is, the one where the detected canopies are distributed as evenly as possible throughout the image. Additionally, the variation coefficient is used instead of the standard deviation in order to be able to compare captures with different ground sample distances or different plantation distances.

This idea is key for the success of the algorithm, specially for the aerial images of regular plantations with thick plant cover on the ground. The principle employed is similar to the approach used by a human to manually discern whether there is a tree canopy or just plant cover in a certain area of the image: the pattern of the plantation is recognised and extrapolated to contiguous areas. If any extrapolated point is included in the area of interest, then the region is probably classified as canopy. If, on the other hand, in that zone there is no extrapolated point, then most likely the area is just some other type of vegetation. From the grower’s point of view, it makes sense ot keep the distances between the olive trees as invariant as possible, since once that the minimum distance is fixed, there would be no reason to increase it, as it would imply reducing plantation density and, consequently, profit.

## 3. Results and Discussion

Generally, during the evaluation of the results of a model, the provided prediction for each input data is compared with the ground truth previously labelled by a person. If the model has a training phase, the input data used during this phase cannot be reused later during the evaluation step. For this, the input data are separated into a training dataset and an evaluation dataset. In addition, to guarantee that the results obtained are independent of the way in which both sets are separated, the evaluation is usually carried out through some type of cross-validation.

For the model proposed in this work, it can be seen (Figure 1) that the type of input data is different for the training phase and the prediction phase. In the first case, pixels labelled as olive tree, ground, weed or shadow are used, while the second case only accepts multispectral captures and their metadata as input. This reality means that the evaluation phase is carried out on complete multispectral captures and not on the pixels that make them up, although the first two blocks (*VC* and *OTCE*) extract the pixels from the captures using the metadata. For this reason, it must be taken into account that the labelled pixels used in the training have to be introduced in a grouped way for each multispectral capture in order to guarantee the selection of those groups that are not used later during the evaluation.

The first step is to carry out the manual labelling process to obtain the ground truth for the training and prediction phases of blocks *VC* and *OTCE*. The labelled regions contain pixels with an absolute certainty of belonging to the chosen class, at the cost of leaving possible pixels that still belong to that class unlabelled (Figure 3). Then, the pixels labelled as olive are used as the ground truth in the rest of the model. This ground truth is a binary mask for each multispectral capture, in which the regions belonging to the olive tree canopies are marked (Figure 10). It is obtained by carrying out the morphological operation of dilation with a circular kernel of 11 pixels in diameter applied to the pixels labelled as olive tree in order to cover the entire region of the olive tree canopies. The resulting masks are then inspected to ensure that this value is correct for all captures.

The following concepts are precised in order to discuss the results of this study:Positive: Set of connected pixels (contour) marked as olive canopy in the ground truth.Negative: Set of connected pixels not marked as olive canopy in the ground truth.Predicted positive: Any coordinate in pixels computed by the model corresponding to an olive tree canopy.Predicted negative: It is not applicable because the model does not compute coordinates where there are no olive tree canopies.True positive: Every predicted positive whose coordinates are within the bounds of some positive.True negative: Not applicable since the predicted negatives do not exist.False positive: Any predicted positive whose coordinates are not within the bounds of any positive.False negative: Any positive for which there is no predicted positive coordinate that is within its bounds.

Table 3 shows the results of performing a 5-fold cross-validation on the model. It shows, from left to right, the ID of the capture (Table 1), the fold in which each capture is part of the test set, the count of positives in the ground truth, the predicted positives, the true positives, the false negatives and the false positives obtained by the model. Finally, three metrics were evaluated: recall, precision and F${}_{1}$ score. In the last row, the sum of each column is calculated, except for the three metrics, which are calculated in the usual way using the previous sums.

For a better graphic interpretation of the results, the contours of the olive tree canopies in the ground truth and the marks of the coordinates predicted by the model are combined on single images. Both contours and marks that are true positives are coloured green, while contours and marks that are false negatives and false positives, respectively, are coloured red. Figure 11 shows an example of these images for 5 of the 18 captures.

Analysing the results table (Table 3), it can be seen that 89% of the captures obtain an F${}_{1}$ score above 0.5, 61% above 0.75, 56% above 0.8 and 28% above 0.9. In 5 of the 18 captures analyzed, practically perfect detection is achieved (Figure 11a,d), in six captures the detection is moderately good (Figure 11b), in five captures the canopies are detected acceptably (Figure 11e) and only in two captures serious detection failures occur (Figure 11c). If the causes of these two failures are analyzed in detail, it is seen that they are not related at all to the main hypothesis proposed in this article—automatic detection of olive tree canopies for groves with thick plant cover is optimised by minimizing the function of coefficients of variation of the lengths of the sides of the Delaunay triangles formed from the coordinates of the canopies predicted by the model. Rather, the causes are generated by the first part of the model, the *VC* in the first instance, classifying shadow pixels as vegetation, and the *OTCE* in the second instance, not being able to exclude them, at least, as weeds. In fact, Figure 11c shows how each and every one of the olive trees is detected, but their shadows are marked instead of the canopies. This is due to the fact that in the probability image the *VC* and the *OTCE* assigned a greater canopy probability to the shadows than to the canopies. The rest of the model from here works well even for images that, due to the time of year they were taken, do not have as much grass on the ground as one would expect.

Table 4 shows the execution times of each model block in the workflow for each capture during the cross-validation, including the *VC* and *OTCE* training time, same at each fold. The last row shows the mean of each column.

As expected, one of the two most unfavorable times is the training time with an average of 57 s per fold. This time must not be taken into account to assess the performance of the model since it is a task that is performed only once. However, another of the most critical times occurs during the prediction of the *OTCM* block with a mean of 57.2 s. Although it may seem like a long time, it is necessary to realise that the time is divided for the 5700 iterations of the brute force method applied in this block, that is, each iteration is executed in approximately one hundredth of a second. Furthermore, this time is no longer a concern considering that, for validating and representation purposes, the number of points computed by the *F* function was higher than it was really needed, and it can be drastically reduced. After these, the next highest time is the prediction time for the *VC* block, taking 32.3 s. This time could only be reduced by finding alternative vegetation classification methods.

Finally, although no articles were found that explicitly deal with the detection or segmentation of tree canopies with thick plant cover on the ground without using three-dimensional data, those that are closely related are shown in Table 5 and Table 6. They summarise information about the article reference, the method used, the publication date, where the data comes from (dataset), what type of data is used (channels), how much data is used (no. of images) and what quality metrics are achieved (accuracy, precision, recall, omission error, commission error and estimation error) in each of them. They are sorted by publication date, and the first row corresponds to the method presented in this article.

## 4. Conclusions

This article presents quite promising results as far as the automatic detection of olive tree canopies is concerned, even more so if we take into account the large amount of plant cover that some of the captures present. This fact makes detection complicated even sometimes for the human eye itself. Considering that olive growing is increasingly oriented towards the search for the quality of the resulting oil, and that organic farming takes advantage of the benefits of maintaining a thick plant cover on the ground, it is concluded that the development of this type of model is an imperative need.

The vast majority of studies carried out with aerial images in the field with olive trees and others fruit trees are based on the search for correlations between the samples collected at the foot of the tree (ground truth) and the features extracted from the images collected by the different types of onboard sensors on the UAV. For this purpose, it is always necessary to extract from the complete images the parts that correspond to the ground truth, usually the canopies. This task receives little attention in scientific works and can be arduous and tedious if performed manually. Even when algorithms are developed to automate this task, they are often used as an aid to the human labeller. In most cases, they work only for groves in which the canopies are clearly delimited by soil with very different spectral features.

The model developed in this article is very useful for performing labelling tasks fully automatically, but due to resulting prediction times of nearly 90 seconds (32.3+0.14+57.2) per capture on average, it cannot be applied in real time for now, that is, the prediction time is greater than the time that elapses between one capture and the next in the multispectral camera. This time, called period, inverse to frames per second (FPS), was set to 1 second during data collection. Despite this, the proposed method has advantages over alternative methods based on structure from motion and multiview stereo to compute 3D information such as photogrammetry or based on other sensors such as *LiDAR*. The first and most obvious one is that it makes predictions from a single capture, while photogrammetry requires a multitude of them in the same area. This fact allows cheaper data collection by reducing the overlap of the captures from 80% to almost 0% without compromising the accuracy. This consequently also reduces the flight time of the UAV. Another advantage is that more complex and costly sensors with *LiDAR* technology or spectrometers that require advanced knowledge and set-up are not used. In fact, only the red, green and blue bands of the multispectral camera were used.

In this way, as future work, it is intended to explore the viability of the model if the input data comes exclusively from a camera whose sensor works in the visible spectrum without radiometric correction. Other possible future research would be to find out the performance of the proposed method applied to other types of groves such as citrus trees, peach trees, chestnut trees or even plants with other typologies such as vines.

The success rate obtained in this study could be improved by evaluating the influence of tree shadows on the *VC* block, increasing the number of images labelled and used for training, or exploring other one-class classification algorithms that may exclude certain types of data. On the other hand, the number of discrete intervals in which each of the three parameters to be optimised is divided could be reduced with the aim of decreasing the prediction time of the block *OTCM*. Furthermore, a more efficient approach could be applied, such as gradient descent, widely used in the back-propagation algorithm of any neural network.

Finally, the proposed model is developed thinking of performing a single task as best as possible. This task is to detect olive tree canopies, that is, to predict with a very high probability the coordinates of the image in which the pixels belonging to those canopies would be found but without applying segmentation. This last task would have to be optimised in the context of a different model. Furthermore, as future work, the region classification task could be optimised, namely, the identification of the plantation from a structured arrangement of trees against regions that would form part of the environment such as roads, forests, buildings, uninhabited land, etc.

## Figures and Tables

**Figure 1 sensors-22-06219-f001:**
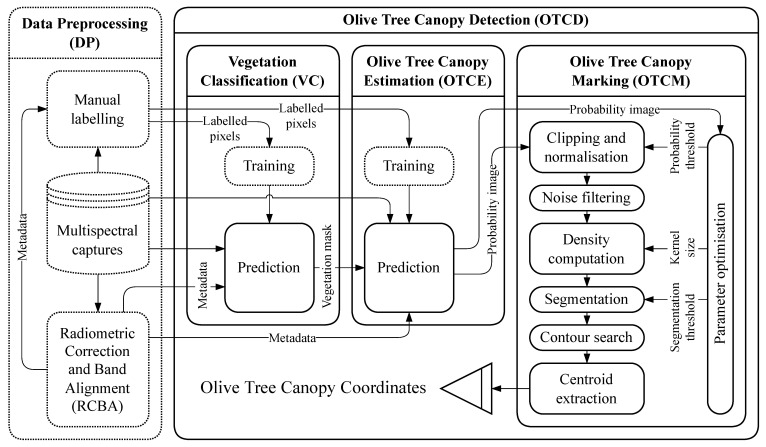
Workflow of the method proposed in this paper. The arrows represent data transfers, and the nodes represent functions that are applied to that data. As exceptions, the initial node, *Multispectral Captures*, represents the data acquired by the UAV and the final node, *Olive Tree Canopy Coordinates*, represents the most optimal coordinates that the model is capable of predicting. Blocks with dotted outlines represent tasks that are performed beforehand.

**Figure 2 sensors-22-06219-f002:**
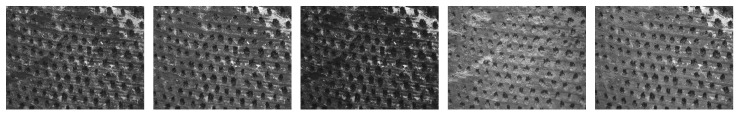
Capture 0, as provided by the multispectral camera: one image for each of the blue, green, red, near-infrared and red-edge bands (**left** to **right**), where the large amount of plant cover in the ground is clearly visible.

**Figure 3 sensors-22-06219-f003:**
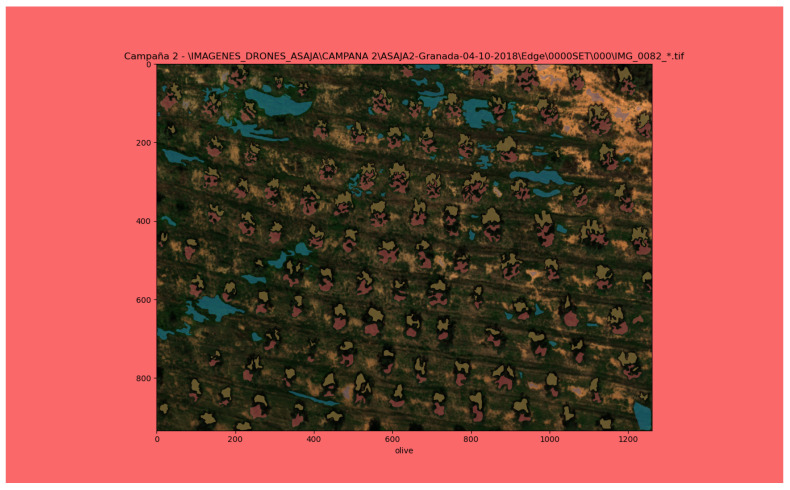
Developed application for the manual labelling of the aerial multispectral images of olive groves. The image shown corresponds to capture 0. The regions are labelled red for the olive tree class, yellow for the shadow class, blue for the weed class and purple for the ground class. The background changes colour to indicate the class selected for labelling.

**Figure 4 sensors-22-06219-f004:**
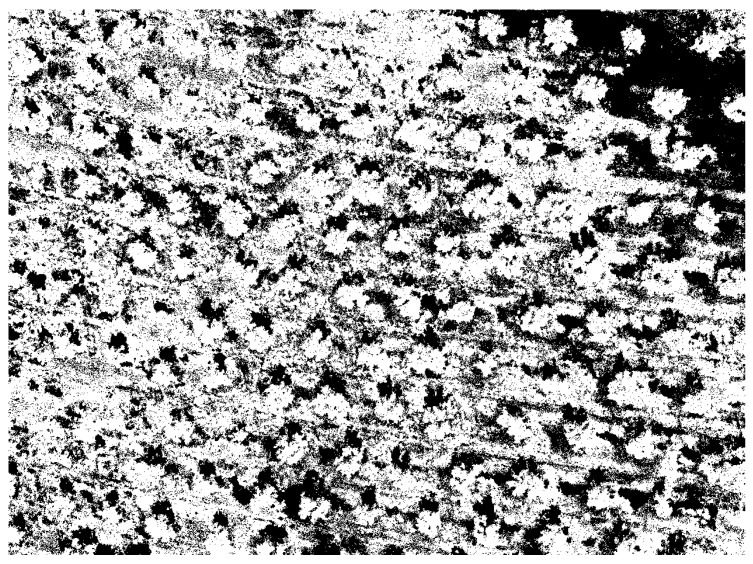
Vegetation mask associated with capture 0. It is a black and white image in which the white pixels were predicted as belonging to the vegetation class by a one-class classification algorithm known as local outlier factor.

**Figure 5 sensors-22-06219-f005:**
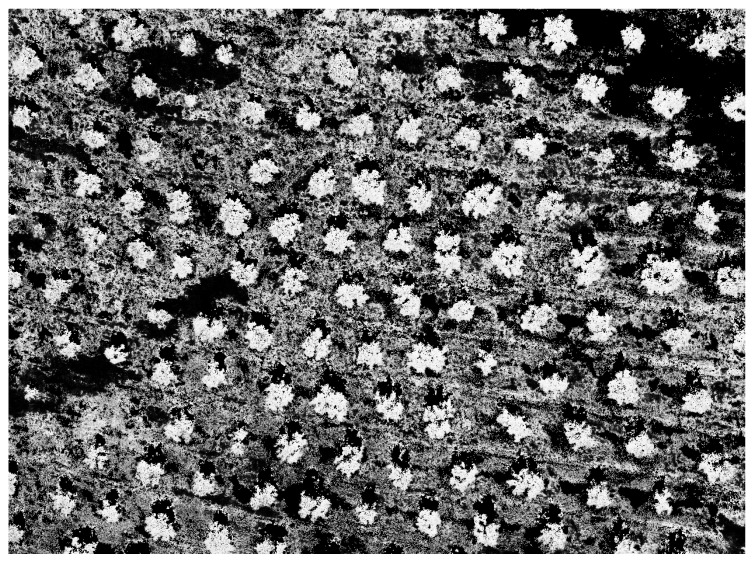
Probability image associated with capture 0. It is a greyscale image in which the values of each pixel indicate the probability of belonging to the olive tree class. This probability is calculated by a decision tree that receives as input only the pixels classified as vegetation during training.

**Figure 6 sensors-22-06219-f006:**
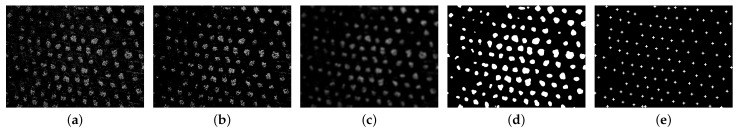
Images resulting from each stage of the *OTCM* block associated with capture 0. The resulting image of each stage is the one received as input by the next stage. (**a**) Clipping and normalisation. (**b**) Noise filtering. (**c**) Density computation. (**d**) Segmentation. (**e**) Centroid extraction.

**Figure 7 sensors-22-06219-f007:**
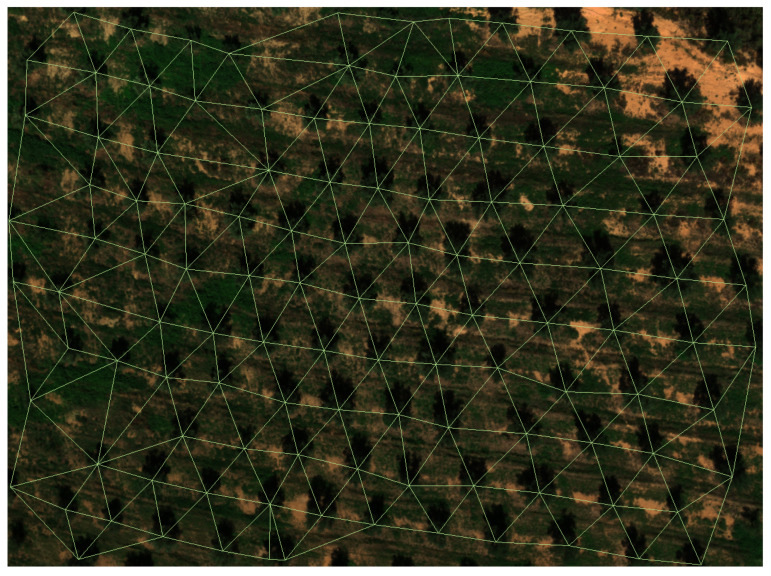
Delaunay triangulation using the coordinates of the olive tree canopies from capture 0, obtained with the optimal combination of parameters calculated by the model whose values are 0.9, 25 and 25 for the probability threshold, the kernel size and the segmentation threshold, respectively.

**Figure 8 sensors-22-06219-f008:**
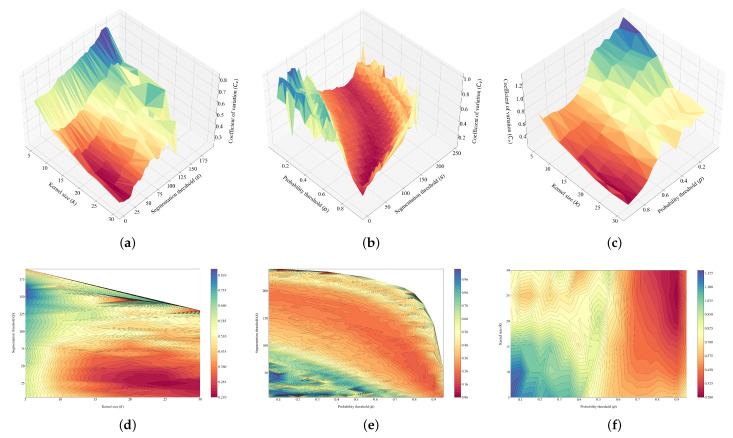
Graphical representation associated with capture 0 in $R^{3}$ of the discrete function *F*, assigning a constant value to one of the parameters in each case. The selected values are the optimum ones computed posteriorly by the model. (**a**) p=0.9. (**b**) k=25. (**c**) s=25. (**d**) p=0.9. (**e**) k=25. (**f**) s=25.

**Figure 9 sensors-22-06219-f009:**
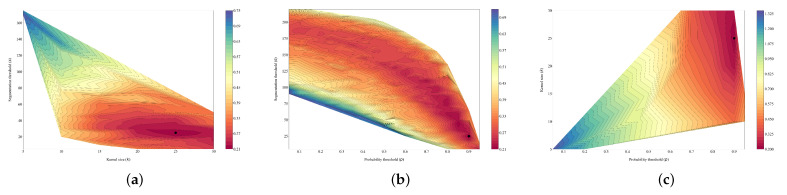
Graphical representation of the redefined function *F* associated with capture 0, fixing one of the parameters in each case. It is observed how the domain of the function changes when it is redefined. The black dot represents the minimum value of the function. (**a**) p=0.9. (**b**) k=25. (**c**) s=25.

**Figure 10 sensors-22-06219-f010:**
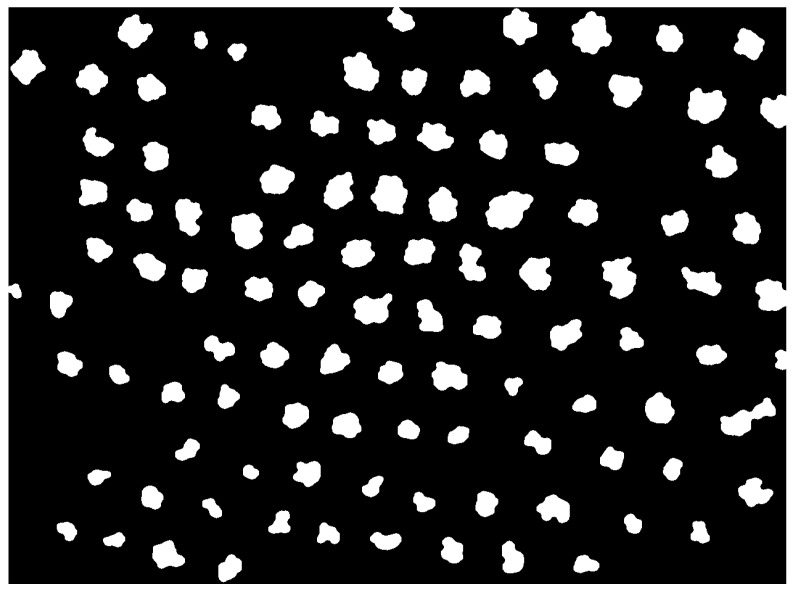
Ground truth associated with capture 0. It is a black and white image in which the white pixels have been manually labeled by a person in order to calculate the quality metrics of the model.

**Figure 11 sensors-22-06219-f011:**
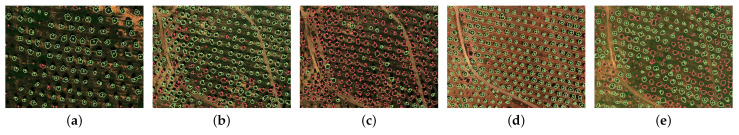
Resulting images obtained and their comparison with the ground truth. Each green mark and contour set indicates a true positive, each red mark indicates a false positive and each red contour indicates a false negative. (**a**) Capture 0. (**b**) Capture 5. (**c**) Capture 8. (**d**) Capture 11. (**e**) Capture 17.

**Table 1 sensors-22-06219-t001:** Detailed characteristics of the 18 captures used to train and test the proposed model. *Campaign* is an identifier of the flight number. For each flight made in different months, 3 captures are selected.

ID	Campaign	Date and Time [ISO8601]	Latitude [DD]	Longitude [DD]	Altitude above Sea Level [m]	Altitude above Ground Level [m]	Ground Sample Distance [cm/px]
**0**	2	2018-10-04T11:00:00Z	37.316932	−3.354593	1408.694	188.774	13.109
**1**		2018-10-04T11:02:56Z	37.316660	−3.354487	1407.542	187.622	13.029
**2**		2018-10-04T11:11:48Z	37.314839	−3.357709	1406.342	186.422	12.946
**3**	3	2018-11-08T11:33:28Z	37.316566	−3.354403	1403.569	180.834	12.558
**4**		2018-11-08T11:33:32Z	37.316642	−3.354134	1403.604	180.869	12.560
**5**		2018-11-08T11:59:06Z	37.314442	−3.354405	1402.180	179.757	12.483
**6**	4	2018-11-29T09:37:17Z	37.316852	−3.354528	1402.846	182.051	12.642
**7**		2018-11-29T09:40:04Z	37.316620	−3.354438	1402.582	181.787	12.624
**8**		2018-11-29T10:18:13Z	37.314431	−3.354456	1401.603	181.069	12.574
**9**	5	2019-07-18T08:13:38Z	37.316618	−3.354430	1409.757	185.211	12.862
**10**		2019-07-18T08:13:44Z	37.316764	−3.353918	1409.754	185.208	12.862
**11**		2019-07-18T09:04:03Z	37.314307	−3.354476	1412.117	189.000	13.125
**12**	6	2019-10-03T08:16:23Z	37.316612	−3.354427	1408.163	183.848	12.767
**13**		2019-10-03T08:17:19Z	37.316682	−3.353719	1409.065	184.750	12.830
**14**		2019-10-03T09:16:04Z	37.314328	−3.354328	1408.841	183.622	12.752
**15**	7	2019-11-07T08:45:24Z	37.316665	−3.354254	1401.534	177.614	12.334
**16**		2019-11-07T08:46:20Z	37.316640	−3.353892	1400.133	176.213	12.237
**17**		2019-11-07T11:09:46Z	37.314307	−3.354477	1403.273	178.476	12.394

**Table 2 sensors-22-06219-t002:** Parameters used to convert raw images to radiance images and their corresponding values and tags extracted from the metadata embedded in the raw images.

Parameter	Metadata Tag	Value
** $t_{e}$ **	ExposureTime	1/988
** *g* **	ISOSpeed/100	200/100
** $DN_{BL}$ **	$(\sum_{i=0}^{4}BlackLevel_{i})/4$	$(\sum_{i=0}^{4}4800)/4$
** $a_{1}$ **	$RadiometricCalibration_{0}$	$1.4511219999999999\times 10^{-4}$
** $a_{2}$ **	$RadiometricCalibration_{1}$	$1.2972460000000001\times 10^{-7}$
** $a_{3}$ **	$RadiometricCalibration_{2}$	$-2.9491650000000001\times 10^{-5}$
** $c_{x}$ **	$VignettingCenter_{0}$	585.34460000000001
** $c_{y}$ **	$VignettingCenter_{1}$	480.0985
** $k_{0}$ **	$VignettingPolynomial_{0}$	$2.049875\times 10^{-7}$
** $k_{1}$ **	$VignettingPolynomial_{1}$	$7.0480179999999998\times 10^{-7}$
** $k_{2}$ **	$VignettingPolynomial_{2}$	$-6.5932030000000001\times 10^{-9}$
** $k_{3}$ **	$VignettingPolynomial_{3}$	$1.907818\times 10^{-11}$
** $k_{4}$ **	$VignettingPolynomial_{4}$	$-2.4725660000000001\times 10^{-14}$
** $k_{5}$ **	$VignettingPolynomial_{5}$	$1.15035\times 10^{-17}$

**Table 3 sensors-22-06219-t003:** 5-fold cross-validation results on the *OTCD* model. In each iteration, the metrics of the model’s predictions are evaluated for the captures with the indicated *IDs*, performing the training with the rest of the captures.

ID	Fold	Positives (P)	Predicted Positives (PP)	True Positives (TP)	False Negatives (FN)	False Positives (FP)	Recall	Precision	F${}_{1}$ Score
2	0	154	166	134	21	32	0.870	0.807	0.838
1		125	141	118	7	23	0.944	0.837	0.887
3		112	115	108	4	7	0.964	0.939	0.952
7		106	98	94	13	4	0.887	0.959	0.922
6	1	100	134	96	4	38	0.960	0.716	0.821
16		100	123	76	26	47	0.760	0.618	0.682
5		238	247	193	45	54	0.811	0.781	0.796
13		103	120	51	52	69	0.495	0.425	0.457
10	2	115	121	110	5	11	0.957	0.909	0.932
12		111	136	90	23	46	0.811	0.662	0.729
15		103	148	67	57	81	0.650	0.453	0.534
0		104	119	101	3	18	0.971	0.849	0.906
8	3	226	238	31	195	207	0.137	0.130	0.134
11		270	285	263	7	22	0.974	0.923	0.948
4		102	119	58	44	61	0.569	0.487	0.525
14	4	263	264	231	32	33	0.878	0.875	0.877
17		225	236	149	80	87	0.662	0.631	0.646
9		123	121	107	16	14	0.870	0.884	0.877
		**2680**	**2931**	**2077**	**634**	**854**	**0.775**	**0.709**	**0.740**

**Table 4 sensors-22-06219-t004:** Execution times of each model block. A training time for each fold and the prediction times for each capture ID are shown. The last row shows the mean of each column.

ID	Fold	VC Training Time (s)	OTCE Training Time (s)	VC Predict Time (s)	OTCE Prediction Time [s]	OTCM Prediction Time (s) for 5700 Iterations
**2**	0	46.3	6.4	28.7	0.15	56.7
**1**				40.4	0.24	65.5
**3**				40.1	0.26	57.7
**7**				34.8	0.13	55.2
**6**	1	57.7	6.4	29.7	0.12	60.1
**16**				46.6	0.16	66.0
**5**				34.8	0.19	59.6
**13**				33.8	0.07	51.0
**10**	2	54.9	7.3	24.5	0.05	43.6
**12**				28.8	0.11	60.9
**15**				28.5	0.11	71.7
**0**				30.8	0.22	61.1
**8**	3	66.4	6.7	28.2	0.14	56.6
**11**				26.1	0.04	52.2
**4**				28.1	0.15	53.1
**14**	4	63.6	6.8	29.2	0.10	50.9
**17**				38.7	0.23	54.3
**9**				29.9	0.09	53.0
		**57.0**	**6.7**	**32.3**	**0.14**	**57.2**

**Table 5 sensors-22-06219-t005:** Comparison of the results of this study with the results of other closely related articles that, although they do not explicitly use images of groves with thick plant cover on the ground, deal with the detection or segmentation of the olive tree canopies. The article reference, the method used, the publication date, where the data comes from (dataset) and what type of data is used (channels) are shown.

Reference	Method	Publication Date	Dataset	Channels
-	Proposed	-	MicaSense RedEdge-MX	Red, green and blue
[48]	Deep learning model (SwinTUnet) based on Unet-like networks	15 January 2022	Satellites Pro	Red, green and blue
[49]	Orthophotos + Mask R-CNN	25 February 2021	Parrot Sequoia camera	Red, green, blue and near infrared
[50]	Edge detection + circular Hough transform	1 June 2020	SIGPAC viewer	Red
[25]	Laplacian of Gaussian + improved k-means clustering	26 February 2020	SIGPAC viewer	Red, green and blue
[51]	Colour-based vs. stereo-vision-based segmentation	5 February 2019	DJI Phantom4 camera	Red, green and blue
[52]	Multi-level thresholding + circular Hough transform	4 December 2018	SIGPAC viewer	Red, green and blue
[53]	Radiometrically corrected orthophotos+ Object-based image analysis	4 December 2017	Modified multiSPEC 4C camera	Red, green, red-edge and near-infrared
[54]	Orthophotos + Thresholding + watershed analysis + microbiological cell counting algorithm	27 October 2017	Leica ADS40, ADS80, ADS100 and DMC III cameras	Red, green, blue and near-infrared
[55]	Optical + radar data + object-based classification	19 July 2011	ADS40 Airborne Digital Sensor + RAMSES + TerraSAR-X satellite	Panchromatic, red, green, blue, near-infrared and X
[56]	K-mean clustering	2 July 2010	SIGPAC viewer	Red, green and blue
[57]	Reticular matching	31 August 2007	Quickbird	Panchromatic

**Table 6 sensors-22-06219-t006:** Comparison of the results of this study with the results of other closely related articles that, although they do not explicitly use images of groves with thick plant cover on the ground, deal with the detection or segmentation of the olive tree canopies. The article reference, how much data is used (no. of images) and what quality metrics are achieved (accuracy, precision, recall, omission error, commission error and estimation error) are shown.

Reference	No. of Images	Accuracy	Precision	Recall	Omission Error Rate	Commission Error Rate	Estimation Error
-	18	N/A	70.9%	77.5%	22.5%	N/A	9.37%
[48]	230	98.3%	N/A	98.8%	1.2%	0.97%	0.94%
[49]	150	N/A	90.07–100%	90.83–100%	0–9.17%	N/A	N/A
[50]	60	N/A	N/A	96%	4%	1.2%	1.27%
[25]	110	97.5%	N/A	99%	1%	4%	0.97%
[51]	10	N/A	99.84%	97.61%	2.39%	N/A	N/A
[52]	N/A	96%	N/A	97%	3%	3%	1.2%
[53]	315	93%	91.6%	95.9%	4.1%	10.57%	4.76%
[54]	4	N/A	N/A	N/A	N/A	N/A	4–27%
[55]	3	76.8–90.5%	N/A	16–66.7%	33.3–84%	0.6–8.4%	N/A
[56]	N/A	N/A	N/A	83.33%	16.67%	0%	
[57]	3	N/A	N/A	93%	7%	5%	1.24%

## Data Availability

The data presented in this study are available on request from the corresponding author. The data are not publicly available due to confidentiality reasons derived from the project members.

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
