# Peer review of "Automatic Detection of Olive Tree Canopies for Groves with Thick Plant Cover on the Ground"

_sensors, 2022, doi:10.3390/s22166219_

Round 1

Reviewer 1 Report

Authors put forward a novel idea of detecting Olive tree canopies with thick plant cover based on relative spatial information on canopies, plant patterns and spectral information. The manuscript have merits to be accepted in the Sensors Journal however there is extensively english editing is required. In abstract section results are missing while introduction section has to be improved by further providing more relevant information regarding tree canopy estimations for other trees as well.

Separate discussion section is required that will throughly discuss the proposed methodology and how we can apply it for other orchards e.g. Citrus. 

Conclusion section is fine but more clarity is needed why we can not apply this method in real time estimation. 

Reviewer 2 Report

1. This study focused on the method for the detection of olive tree canopies from high-resolution aerial images containing information in the visible spectrum, regardless of the plant cover density between canopies. The methodology makes sense. I am suggesting a potential publication of this research after thorough minor review.

2. Please modify and remove unnecessary keywords.

3. The abstract should contain important keywords for searching. The abstract lacks the result of the method at the end, please add.

4. The authors are urged to add some reference on the use of UAV data to detect crops automatically the following can be added:

Gao, M.; Yang, F.; Wei, H.; Liu, X. Individual Maize Location and Height Estimation in Field from UAV-Borne LiDAR and RGB Images. Remote Sens. 2022, 14(10), 2292. https:// doi.org/10.3390/rs14102292

Xu, W.; Deng, S.; Liang, D.; Cheng, X. A Crown Morphology-Based Approach to Individual Tree Detection in Subtropical Mixed Broadleaf Urban Forests Using UAV LiDAR Data. Remote Sens. 2021, 13, 1278. https:// doi.org/10.3390/rs13071278

5.Line61-62: There should add some sentences and references to highlight the innovations of this study. Only line 61-64 is poor.

6. The introduction section lacks recent advances in research on olive tree canopy detection.

7.Figure1: The text of the connection line is too small, please fix it.

8.Line160: Please add more descriptions or values of all the parameters included in Equations (1) to (5)

9. Check line 24 “of wich 57.9%”

10. Table 3: Please check the orientation of the first line of text.

11. The Discussion section need to be significantly improved. Presently, it contains only a brief discussion solely from this study. The point of discussion is that you compare it to previous research and describe the implications and scientific contribution of this study.

Reviewer 3 Report

The objective of this paper is to develop a method to segment olive tree canopies from very high-resolution aerial images where there is a high level of plant cover in the ground between canopies. Although there is some novelty in the proposed algorithm (in the way of refining the spectral information to achieve better crown classification), the authors completely neglect the possibility of extracting structural information from the images (i.e., using photogrammetric analysis by acquiring images with large swath overlap). Also, olive trees are very easy to detect as they are regularly planted with no/little overlap between crowns (if we combine structural information). Overall, the paper adds little value to advancing the state-of-the-art in tree crown delineation. The authors should develop a method that exploits both spectral and spatial information in the data to perform crown delineation and compare it with other state-of-the-art methods to get an idea of the relative performance of the proposed methods.

Additional comments:

The title "Automatic Detection of Olive Tree Canopies for Crops with Thick Plant Cover on the Ground", is not grammatically correct! What do you mean by Detecting Olive Tree Canopies for Crops?

Line 113 "This way, a set of 18 captures (Table 1) with a large amount of plant cover in the ground, taken from an olive grove in the town of Diezma, province of Granada, in Andalucia, Southern Spain." - This sentence is incomplete.

All the figure labels are too short and do not allow one to have a complete understanding of the figure.

Reviewer 4 Report

In this study, the authors develop a standard method for the detection of olive tree canopies from high-resolution aerial images containing information in the visible spectrum, regardless of the plant cover density between canopies. Experimental results show the good performance of the proposed method. However, some issues should be addressed.

Major issues:

1) Although the authors test the experiments for olive tree canopies. But I think that the method proposed by the authors is universal. In other words, can the authors verify the effectiveness of the proposed method in other trees? It is recommended to add at least one more set of experimental data.

2) Is there any other way to get results for the detection of olive tree canopies. Can the authors compare the proposed method with other similar methods?

Minor issues:

1) The methods proposed by the authors is mainly based on remote sensing image. In the introduction part, some other remote sensing image processing methods should also be introduced, e.g.,

[1] Super-Resolution Mapping Based on Spatial-Spectral Correlation for Spectral Imagery [J]. IEEE Transactions on Geoscience and Remote Sensing, 2021, 59(3): 2256-2268.

[2] A Simple Method of Mapping Landslides Runout Zones Considering Kinematic Uncertainties, Remote Sensing, 2022, 14(3): 668.

2) The last part of the introduction needs to give chapter arrangement. In addition, there are some grammatical errors in the article, which need further careful proofreading.
